# COVID-19 Vaccine Booster Hesitancy (VBH) and Its Drivers in Algeria: National Cross-Sectional Survey-Based Study

**DOI:** 10.3390/vaccines10040621

**Published:** 2022-04-15

**Authors:** Mohamed Lounis, Djihad Bencherit, Mohammed Amir Rais, Abanoub Riad

**Affiliations:** 1Department of Agro-Veterinary Science, Faculty of Natural and Life Sciences, University of Ziane Achour, Djelfa 17000, Algeria; 2Department of Biology, Faculty of Natural and Life Sciences, University of Ziane Achour, Djelfa 17000, Algeria; djihadbencherit88@gmail.com; 3Department of Dentistry, Faculty of Medicine, University of Algiers Benyoucef Benkhedda, Algiers 16000, Algeria; ma.rais@univ-alger.dz; 4Department of Public Health, Faculty of Medicine, Masaryk University, 625 00 Brno, Czech Republic; abanoub.riad@med.muni.cz

**Keywords:** COVID-19, vaccine, vaccine booster, vaccine hesitancy, vaccine acceptance, Algeria

## Abstract

Due to the emergence of various highly contagious variants of SARS-CoV-2, vaccine boosters were adopted as a complementary strategy in different countries. This strategy has, however, posed another challenge for the national authorities to convince their population to receive the booster after the first challenge of COVID-19 primer dose vaccines. This study was conducted to determine COVID-19 vaccine booster acceptance and its associated factors in the general population in Algeria. Using social media platforms, an online self-administered questionnaire was distributed between 28 January and 5 March 2022 for all Algerian citizens who received COVID-19 vaccines. Overall, 787 respondents were included in this study. Among them, 51.6%, 25%, and 23.8% accepted, rejected, or were hesitant about the COVID-19 vaccine booster, respectively. However, only 13.2% declared receiving the booster dose. Additionally, while 58.2% of the respondents declared being relieved after primer vaccination, 11.4% among them declared that they regretted being vaccinated. The most common reasons for acceptance were experts’ recommendations (24.6%) and the belief that COVID-19 vaccine boosters were necessary and efficient, while rejection was mainly due to the belief that primer doses are sufficient (15.5%), or that vaccination in general is inefficient (8%). Males, older individuals, those with chronic comorbidities or a history of COVID-19 infection, non-healthcare workers, and those with low educational levels were associated with significantly higher odds for booster acceptance. Moreover, belief that booster doses were necessary and efficient, disagreeing with the notion that primer doses were not sufficient, experts’ recommendations, and the desire to travel abroad were significantly associated with higher odds of COVID-19 vaccine booster acceptance.

## 1. Introduction

The world is still in battle against the coronavirus disease (COVID-19) pandemic since its start two years ago, with a heavy balance of four hundred and seventy-six million cases and more than six million deaths [1].

After trying multiple non-pharmaceutical measures including border closure, social distancing measures and containments, which achieved a relative slowing of the SARS-CoV-2 dissemination, countries and international health authorities realized that these measures were not sufficient to completely control the disease [2,3]. Later, they opted for another strategy known as “herd immunity or population immunity” [4]. This approach refers to the indirect protection from an infectious disease occurring after immunization of a large portion of a community either through vaccination or immunity developed through previous infection. It will limit the spread of the disease, and consequently, the whole community becomes protected—not just those who are immune [5]. The best approach to achieve “herd immunity” recommended by the World Health Organization (WHO) is to protect people by vaccination [6].

The proportion of the population that must be vaccinated to achieve herd immunity with vaccination is not well-defined. It will vary according to multiple factors such as the community, the vaccine and the population prioritized for vaccination. A study by Anderson et al., 2020 showed that herd immunity could be achieved with 75–90% vaccination coverage for 100% vaccine efficacy with long-term protection and the entire population for lower efficacies [7].

Since the emergency authorization of some COVID-19 vaccines in December 2020, ten vaccines are currently approved by the WHO, and 33 COVID-19 vaccines are approved by at least one country [8].

Despite misinformation and conspiracy theories surrounding COVID-19 vaccines that have highly influenced vaccine uptake, more than 10.16 billion doses have been administered worldwide [9]. In addition, after high rates of vaccine hesitancy reported in the first months in different countries, results obtained in the last months demonstrated that people’s attitudes trended more frequently toward vaccine acceptance [10,11,12,13,14,15,16].

However, the emergence of different variants of SARS-CoV-2 and the relative decreases in vaccine efficacy against them have created a real dilemma. While many countries have opted for the administration of a vaccine booster to increase the protection of vaccinated people, this strategy did not gain a real consensus among scientists [17].

Consequently, this strategy is again faced with the challenge of vaccine booster hesitancy or rejection. For instance, results obtained in developed countries are encouraging, with rates of willingness to receive vaccine boosters varying from 61.8% to 95.5% in the USA, Poland, Czech Republic, Germany, Japan, China and Denmark [18,19,20,21,22,23,24,25,26].

Data from low- and lower-middle-income countries are, however, scarce. Algeria is one of these countries. It is the eleventh most affected African country with a total of 265,694 infections and 6419 deaths [1]. Based mainly on four COVID-19 vaccines including Sputnik V, AstraZeneca, Sinopharm and Sinovac vaccines, the vaccination campaign in the country was initiated in January 2021 [8]. However, this campaign was characterized by a high rate of vaccine hesitancy, as confirmed by either vaccination statistics (only 13.7% are fully vaccinated) [9] and published studies [27,28,29,30,31]. Regarding the COVID-19 vaccine booster, the current data show that 490,676 (1.1% of the vaccinated population) persons have received it since its start in November 2021 [32]. Therefore, the present work was conducted to evaluate vaccine booster hesitancy and acceptance in Algeria’s vaccinated population and to determine the associated demographic, anamnestic and psychosocial factors.

## 2. Materials and Methods

### 2.1. Design

A descriptive cross-sectional survey-based study was carried out between 28 January and 5 March 2022 in accordance with the Strengthening the Reporting of Observational Studies in Epidemiology (STROBE) guidelines for cross-sectional studies [33]. To collect data from the target population, the study utilized a self-administered questionnaire (SAQ) that was developed and disseminated online through Google Forms (Google LLC, Menlo Park, CA, USA, 2021).

### 2.2. Population

The target population of this study was the adult population in Algeria, from which participants were recruited using a non-random sampling technique through snowballing recruitment. The eligibility criteria included: (i) being an Algerian national at least 18 years old, (ii) capacity to communicate in Arabic or French, and (iii) being previously vaccinated against SARS-CoV-2. Potential participants were invited to access the digital SAQ through either a uniform resource locator (URL) or a quick response (QR) code. Using social media platforms, participants were invited to complete the questionnaire voluntarily without any incentives.

The minimum sample size was computed using Epi-Info^TM^ version 7.2.4 (CDC. Atlanta, GA, USA, 2020), with the following assumptions: confidence level 95%, expected COVID-19 vaccine booster acceptance 50%, error margin 4%, and postulated proportion of invalid responses generated by careless/insufficient effort (C/IE) 10% [34]. The pragmatic sample size for this study was 660 responses.

Participation in this study was not incentivized by any means of reward, and participants’ interest in taking part was not coerced by any means or threat. Participants’ identities were kept anonymous to control for the Hawthorne effect and information bias.

A total of 790 responses were initially received, out of which three responses were excluded due to a lack of information about their attitudes towards COVID-19 vaccine booster doses.

### 2.3. Instrument

The SAQ used in this study was developed based on previous literature about COVID-19 vaccine booster hesitancy [18,19,20,21,22]. It had twenty-seven items that were predominantly close-ended and stratified into four sections: (i) demographic characteristics including sex, age group, profession, educational level, and region, (ii) medical anamnesis including chronic illnesses and influenza vaccine, (iii) COVID-19-related anamnesis including previous infection, onset, severity, vaccine type, and post-vaccination relief, regret, and preventive measures, and (iv) attitudes towards COVID-19 vaccine booster doses and promoters of and barriers to their acceptance.

The dependent variable was the willingness to receive COVID-19 vaccine booster doses stratified into three levels (rejection, hesitancy, acceptance). The independent variables included demographic and anamnestic characteristics and psychological perceptions. Content validity of the instrument was reviewed by a panel of experts who verified its content and provided guiding comments that helped develop the final version. The digital SAQ was provided in bilingual format (Arabic and French).

### 2.4. Ethics

The study protocol was reviewed and approved by the Scientific Committee of the Faculty of Natural and Life Sciences, the University of Djelfa, on 25 January 2022. All participants provided their informed consent digitally prior to their participation, and they were allowed to withdraw from the study at any moment without the need to justify their decision. The Declaration of Helsinki for research involving human subjects and the general data protection regulation (GDPR) were followed during the design and implementation of this study [35,36]. No identifying personal information was collected from the participants; therefore, retrospective identification of the participants was not possible.

### 2.5. Analyses

The Statistical Package for the Social Sciences (SPSS) version 28.0 (SPSS Inc. Chicago, IL, USA, 2021) and the R-based open software Jamovi were used to perform all statistical analyses [37,38]. Initially, descriptive statistics were performed to present the qualitative variables using frequencies (*n*) and percentages (%). Inferential statistics were performed using the Chi-squared test (χ^2^), Fisher’s exact test for <5 cases, and binary logistic regression to test the association between independent and dependent variables. Subsequently, multivariate logistic regression was performed to compute the adjusted odds ratio (AOR) of various psychosocial factors for COVID-19 vaccine booster dose acceptance. All analytical tests were performed following the assumptions of a confidence level of 95% and significance level of <0.05.

## 3. Results

### 3.1. Demographic Characteristics

Out of the 787 included participants, 61.6% were females, and 61.2% were non-healthcare professionals. Females were significantly (*p* < 0.001) more represented in the healthcare professionals’ group (70.2%) than in the non-healthcare professionals’ group (56.2%). The most represented age group was 31–40 years old (31.3%), followed by 18–30 years old (26.9%) and 41–50 years old (24.4%). Most participants were married (61.1%) without a statistically significant difference (*p* = 0.590) between healthcare professionals (62.3%) and non-healthcare professionals (60.4%). The vast majority were living in urban areas (91.2%) without a statistically significant difference (*p* = 0.478) between healthcare professionals (92.1%) and non-healthcare professionals (90.7%). Only 8.1% of the participants had a college or school level of education, which was more common among non-healthcare professionals (10.4%) than healthcare professionals (4.6%). Less than half (48.2%) of the sample reported having postgraduate degrees (Master’s or above), which were more common among healthcare professionals (55.7%) than non-healthcare professionals (43.4%) (Table 1).

### 3.2. Anamnestic Characteristics

On evaluating their medical anamnesis, 27.8% of the participants reported having at least one chronic illness without a statistically significant difference (*p* = 0.983) between healthcare professionals (27.9%) and non-healthcare professionals (27.8%). The most commonly reported disease was chronic hypertension (8.9%), followed by diabetes mellitus (8%) and respiratory diseases (5.6%). Less than one quarter (24.3%) of the participants reported receiving the influenza vaccine recently without a statistically significant difference (*p* = 0.089) between healthcare professionals (27.5%) and non-healthcare professionals (22.2%) (Table 2).

On evaluating their COVID-19-related anamnesis, less than two-thirds (65.3%) of the sample reported being previously infected by SARS-CoV-2, with healthcare professionals (75.1%) being significantly (*p* < 0.001) more frequently infected than non-healthcare professionals (59.1%). More than half (51.4%) of those previously infected were infected before receiving the first vaccine dose, followed by 39% infected after the second dose. Only 2.5% were infected after the third dose, and 4.5% reported hospitalization due to SARS-CoV-2 infection. While the vast majority (90.6%) reported having at least one COVID-19 infection among their family, 41.9% reported that one of their family members passed away due to COVID-19 (Table 3).

On evaluating their COVID-19 vaccine-related anamnesis, the most commonly administered vaccine type was Sinovac (66.1%), followed by AstraZeneca-Oxford (12.6%) and Sputnik V (10.2%). The least common vaccine was Pfizer-BioNTech, which was received by only 4 participants, representing 0.5% of the sample. While inactivated virus vaccines were more common among non-healthcare professionals (77.8%) than healthcare professionals (65.9%), adenoviral vector vaccines were significantly more common among healthcare professionals (33.4%) than non-healthcare professionals (21.8%) (*p* < 0.001). Most participants reported following preventive measures even after being vaccinated (88.8%).

While 58.2% of the participants felt relieved after vaccination, only 11.4% regretted being vaccinated. The most common reason for regretting vaccination was being infected after receiving the COVID-19 vaccine (breakthrough infection) (6.9%), followed by the belief that vaccines are inefficient (6.7%) and being burdened by post-vaccination side effects (3%). Only 13.2% of the participants reported receiving the third dose by the time they responded to the questionnaire, and the most commonly administered vaccine type for booster doses was Sinovac (48%), followed by Janssen (15.2%) and AstraZeneca-Oxford (12.8%) (Table 4).

### 3.3. COVID-19 Vaccine Booster-Related Attitudes

Overall, more than half (51.6%) of the participants indicated their acceptance to receive COVID-19 vaccine booster doses, while one quarter (25%) rejected booster doses, and 23.4% were hesitant. Healthcare professionals were significantly (*p* = 0.011) less inclined to accept (45.9%) booster doses than non-healthcare professionals (55.2%).

The most common reason for acceptance was experts’ recommendation (24.6%), which was more common for non-healthcare professionals (29.3%) than healthcare professionals (17%), followed by the belief that COVID-19 vaccine boosters were necessary and efficient (23.4%). The most preferred COVID-19 vaccine type was Sinovac (33.3%), followed by Janssen (12.6%), AstraZeneca-Oxford (11.8%), and Pfizer-BioNTech (9.6%). The most common reason for rejection was the belief that primer doses are sufficient (15.5%), followed by the belief that vaccination, in general, is inefficient (8%), the belief that vaccines could impact the immune system adversely (6.5%), and the fear of post-vaccination side effects (5%) (Table 5).

### 3.4. Promoters of and Barriers to COVID-19 Vaccine Booster Acceptance

Male participants had a significantly (*p* < 0.001) higher acceptance level (59.9%) than females (46.4%). Similarly, the older age groups, i.e., those aged >60 years and between 51–60 years, had significantly (*p* < 0.001 and = 0.002) higher acceptance levels (71.8% and 66.3%) than the group aged 18–30 years (43.9%), respectively. There were no statistically significant differences between single (49.3%) versus married (53%) participants and urban (51.3%) versus rural residents (55.1%) in terms of COVID-19 vaccine booster dose acceptance. Individuals with low educational levels (college or school) had the highest acceptance level (71.9%) compared with bachelor’s degree holders (47.1%) and postgraduate degree holders (52.2%) (Table 6).

### 3.5. Regression Analyses

Primarily, binary logistic regression was performed to evaluate the odds ratios (OR) of demographic and anamnestic factors for COVID-19 vaccine booster acceptance based on significant results from the previous analyses. Males (OR: 1.729; CI 95%: 1.292–2.313), the age group of >60 years old (OR: 3.257; CI 95%: 1.541–6.884), and those suffering from chronic illness (OR: 1.394; CI 95%: 1.018–1.908) had significantly higher odds for COVID-19 vaccine booster acceptance. The highest odds ratio for acceptance was among those who felt post-vaccination relief (OR: 8.120; CI 95%: 4.892–13.479). Contrarily, healthcare professionals (OR: 0.689; CI 95%: 0.517–0.919) and those with postgraduate degrees (OR: 0.428; CI 95%: 0.239–0.765) had lower odds of acceptance (Table 7).

Subsequently, multivariable logistic regression analysis adjusted for sex, age group, educational level, profession, chronic illness, previous COVID-19 infection, post-vaccination relief and regret was performed to estimate the adjusted odds ratio (AOR) of COVID-19 vaccine acceptance. The highest AOR was among those participants who believed that booster doses were necessary and efficient (AOR: 28.112; CI 95%: 13.235–59.709), followed by disagreement with the notion that primer doses were not sufficient (AOR: 23.641; CI 95%: 11.087–50.409), and having no breakthrough infections (AOR: 6.870; CI 95%: 0.783–60.248). Fear of side effects was not statistically significant, as well as breakthrough infections. Experts’ recommendations (AOR: 4.801; CI 95%: 3.116–7.398) and the desire to travel abroad (AOR: 1.804; CI 95%: 1.136–2.863) were significant promoters of COVID-19 vaccine booster acceptance (Table 8).

## 4. Discussion

The main objective of this study was to determine the attitudes and associated factors related to COVID-19 vaccine booster acceptance and hesitancy among the general public in Algeria. To the best of the authors’ knowledge, this is the first study regarding the COVID-19 vaccine booster in Algeria. In this way, after the challenge of convincing the Algerian public to receive the COVID-19 vaccine when vaccine acceptance did not exceed 51.1% [31], the new task for the Algerian health authorities has been to encourage vaccinated individuals to receive the vaccine booster after it became available in November 2021.

The results of our study revealed that 51.6% of all respondents were in favor of the COVID-19 vaccine booster, while 25% rejected it. Moreover, only 13.2% of all respondents declared that they had received the COVID-19 vaccine booster. However, this rate is higher than the current rate in Algeria, estimated at 1% (436,274 administered doses) [32]. The rate of acceptance identified in this study is not far from the reported rate of 55% in Saudi Arabia [39] but higher than the reported rate in Jordan (39%) [40]. Globally, vaccine booster acceptance is generally higher in developed countries. Some available data showed rates of 79.1% and 83.6% among adults and healthcare workers (HCWs) in the United States [18,19], 67.4% and 71% in Poland [20,41], 71.3% among HCWs in Czechia [21], 84.5% among medical students in Japan [23], 85.7% in the university community in Italy [42], 84.8% and 93.7% among the general population in China [25,43], 87.8% among university academics and students in Germany, and 95.5% among the adult Danish population [26].

The most common reasons for acceptance in this study were experts’ recommendations (24.6%), followed by the belief that COVID-19 vaccine boosters were necessary and efficient (23.4%), while the most common causes for rejection were the belief that primer doses were sufficient (15.5%), the belief that vaccination—in general—is inefficient (8%) or could impact immune system (6.5%), and the fear of post-vaccination side effects (5%). In this way, previous studies reported that one of the most common causes of vaccine booster rejection was the presence of side effects after the primer doses [41]. In the same direction, post-vaccination relief greatly increased the odds of acceptance of vaccine boosters (OR: 8.120; CI 95%: 4.892–13.479).

It is known that elderly people are one of the prioritized categories for COVID-19 vaccine boosters [44]. Fortunately, in our study, aged persons were more likely to accept vaccine boosters than younger ones (OR: 3.257; CI 95%: 1.541–6.884). The same results were also obtained for individuals with chronic diseases (OR: 1.394; CI 95%: 1.018–1.908) which is in accordance with the results of previous studies reporting that prioritized groups were more likely to accept vaccine boosters [20,43,45]. These results could be very helpful for health authorities, given the high risk of COVID-19 infection in these categories. Moreover, previous infection with COVID-19 was associated with higher odds of acceptance of a vaccine booster than among those who were not in contact with this disease in this study.

The surprising result for the prioritized groups is, however, in relation with healthcare workers’ attitudes. In fact, healthcare workers had lower odds of acceptance of a vaccine booster dose than the general population (OR: 0.689; CI 95%: 0.517–0.919). Hence, hesitancy among healthcare workers could have a negative effect for the general public, given their role as the main sources of medical information for the general population. This point should be explored to better understand the reasons for vaccine booster hesitancy among members of this category. Globally, the healthcare profession was generally associated with high acceptance of either primers or booster doses of COVID-19 vaccine [18,21,41,43]. In a previous study of COVID-9 vaccine acceptance (primer doses) in Algeria, healthcare workers were more likely to accept this vaccine than non-health workers [31].

Similarly, our results showed that the higher the educational level, the lower the acceptance rate. In fact, individuals with university level (OR: 0.348; CI 95%: 0.194–0.625) and postgraduate degrees (OR: 0.428; CI 95%: 0.239–0.765) had lower odds of acceptance than those with a college/school level education. These results did not agree with previously published studies in different countries.

For gender, the question seems to become increasingly clear with males being more in favor of vaccines in general than females. As in our results (OR: 1.729; CI 95%: 1.292–2.313), male acceptance was previously confirmed for COVID-19 vaccines primers [12,28,46] and booster doses [19,47,48]. The most reported explanations of these low rates of acceptance among females are related to their psychological and hormonal characteristics [20].

Apart from these socio-demographic and anamnestic factors, we also assessed the influence of behavioral and psychological factors on vaccine acceptance. Results showed that the belief that booster doses were necessary and efficient (AOR: 28.112; CI 95%: 13.235–59.709), and the disagreement with the notion that primer doses were not sufficient (AOR: 23.641; CI 95%: 11.087–50.409) were highly associated with booster acceptance. Other promoters of COVID-19 vaccine booster acceptance included experts and scientists’ recommendations (AOR: 4.801; CI 95%: 3.116–7.398) and the desire to travel (AOR: 1.804; CI 95%: 1.136–2.863), while fear of side effects and breakthrough infections were not statistically associated with COVID-19 booster acceptance. These findings join results of previous studies that suggested that confidence in vaccine safety and effectiveness and trust in pharmaceutical companies would enhance vaccine acceptance rates and thus help to increase the number of vaccinated people [47,48,49]. These observations indicate the important role of public health education in increasing public vaccine acceptance.

In the population study, the most commonly administered vaccine type was Sinovac (66.1%), followed by AstraZeneca (12.6%) and Sputnik V (10.2%). However, none of the vaccine types or technologies affected attitudes toward booster doses. Booster preference closely mirrored that for the Sinovac (33.3%), Janssen (12.6%), AstraZeneca (11.8%), Pfizer-BioNTech (9.6%) and Sputnik V (7.1%) vaccines. These findings showed that vaccinated people are more likely to prefer a booster dose from the same vaccine. However, as reported by Rzymski et al., 2021 [20] and Alhasan et al., 2021 [39], vaccinated individuals do not necessarily prefer the same vaccine that they received in the primer doses. In the same way, the last authors reported that persons who believe that combining different vaccines is effective against variants are significantly more likely to accept a vaccine booster. The preference for inactivated vaccines could be related to the conventional production method and their lower side effects compared to the other vaccines, as confirmed in previous studies. Additionally, preferences for the other vaccines, such as those from Janssen, AstraZeneca-Oxford, and Pfizer-BioNTech, could be mainly related to trust in the manufacturing company.

### 4.1. Limitations

This study had some limitations related essentially to the sampling method. We used a non-random sampling technique through snowballing recruitment on a generated selection basis, which could affect the generalization of the results to the entire Algerian population. The online survey method used in this study could have excluded people with no access to the internet and those who could not read or write. Younger persons who spend more time with social media were more likely to be covered than other groups. The cross-sectional nature of this study could only provide a snapshot of acceptance and/or hesitancy, which could change through time. Additionally, booster dose acceptance and hesitancy were assessed by self-reporting, and we did not use a scale standard of acceptance and hesitancy. Furthermore, the survey did not include information about side effects of primer doses of COVID-19 vaccines. Finally, the response rate could not be calculated due to the impossibility of determining the total number of people reached by the survey.

### 4.2. Strengths

This is the first study of the general population’s attitudes toward the COVID-19 vaccine booster and factors associated with its acceptance in Algeria. Limited data are available in developing countries on this question. The study was conducted using an online survey that allowed the respondents to complete the questionnaire in a private environment, thus reducing some social and interviewer biases. In addition, though not representative, the sample covered almost all Algerian sectors and categories related to the socio-demographic and anamnestic characteristics of the population (age, sex, marital status, residence, chronic illnesses, COVID-19 infection, etc.).

The results of this study could be helpful for health authorities in their campaign to promote COVID-19 booster awareness and uptake. These results could be of great importance for African and other developing countries where vaccine hesitancy was generally reported at high levels. They could be a key in understanding this phenomenon and contribute to its mapping around the world.

## 5. Conclusions

In conclusion, this study reported for the first time the attitudes of the Algerian public toward the COVID-19 vaccine booster. A medium rate of booster acceptance was obtained, which remains lower than those reported in developed countries.

The study identified specific groups with the highest rates of booster acceptance (males, persons over the age of 60 years, and those suffering from chronic diseases) and hesitancy (healthcare professionals and postgraduate students). It also described that vaccine booster acceptance is mainly related to certain psychological factors related to vaccine effectiveness and safety. Taking these factors into consideration could be helpful for the national authorities in their effort to convince the general population of the importance of the COVID-19 vaccine booster in the battle against COVID-19.

## Figures and Tables

**Table 1 vaccines-10-00621-t001:** Demographic Characteristics of Algerian Adults Participating in COVID-19 VBH Survey (*n* = 787).

Variable	Outcome	Non-Healthcare Professionals (*n* = 482)	Healthcare Professionals (*n* = 305)	Total (*n* = 787)	*Sig.*
Sex	Female	271 (56.2%)	214 (70.2%)	485 (61.6%)	**<0.001**
	Male	211 (43.8%)	91 (29.8%)	302 (38.4%)	
Age Group	18–30 years old	127 (26.3%)	85 (27.9%)	212 (26.9%)	0.640
	31–40 years old	139 (28.8%)	107 (35.1%)	246 (31.3%)	0.066
	41–50 years old	125 (25.9%)	67 (22.0%)	192 (24.4%)	0.207
	51–60 years old	65 (13.5%)	33 (10.8%)	98 (12.5%)	0.270
	>60 years old	26 (5.4%)	13 (4.3%)	39 (5%)	0.476
Marital Status	Single	191 (39.6%)	115 (37.7%)	306 (38.9%)	0.590
	Married	291 (60.4%)	190 (62.3%)	481 (61.1%)	
Residence	Urban	437 (90.7%)	281 (92.1%)	718 (91.2%)	0.478
	Rural	45 (9.3%)	24 (7.9%)	69 (8.8%)	
Educational Level	College/School	50 (10.4%)	14 (4.6%)	64 (8.1%)	**0.004**
	Bachelor’s Degree	223 (46.3%)	121 (39.7%)	344 (43.7%)	0.069
	Masters’ Degree or above	209 (43.4%)	170 (55.7%)	379 (48.2%)	**<0.001**

Chi-squared test (*χ*^2^) was used with a significance level *Sig.* < 0.05. Statistically significant differences are indicated with bold character.

**Table 2 vaccines-10-00621-t002:** Medical Anamnesis of Algerian Adults Participating in COVID-19 VBH Survey (*n* = 787).

Variable	Outcome	Non-Healthcare Professionals (*n* = 482)	Healthcare Professionals (*n* = 305)	Total (*n* = 787)	*Sig.*
Chronic Illness	Diabetes Mellitus	35 (7.3%)	28 (9.2%)	63 (8%)	0.334
	Chronic Hypertension	39 (8.1%)	31 (10.2%)	70 (8.9%)	0.320
	Cardiovascular Disease	8 (1.7%)	7 (2.3%)	15 (1.9%)	0.525
	Respiratory Disease	29 (6%)	15 (4.9%)	44 (5.6%)	0.513
	Renal Disease	2 (0.4%)	2 (0.7%)	4 (0.5%)	0.643 *
	Other	62 (12.9%)	34 (11.1%)	96 (12.2%)	0.474
	Total	134 (27.8%)	85 (27.9%)	219 (27.8%)	0.983
Influenza Vaccine	No	375 (77.8%)	221 (72.5%)	596 (75.7%)	0.089
	Yes	107 (22.2%)	84 (27.5%)	191 (24.3%)	

Chi-squared test (*χ*^2^) and Fisher’s exact test (*) were used with a significance level *Sig.* < 0.05.

**Table 3 vaccines-10-00621-t003:** COVID-19 Infection-related Anamnesis of Algerian Adults Participating in COVID-19 VBH Survey (*n* = 787).

Variable	Outcome	Non-Healthcare Professionals (*n* = 482)	Healthcare Professionals (*n* = 305)	Total (*n* = 787)	*Sig.*
COVID-19 Infection	No	197 (40.9%)	76 (24.9%)	273 (34.7%)	**<0.001**
	Yes ^+^	285 (59.1%)	229 (75.1%)	514 (65.3%)	
^+^ Onset	Before 1st Dose	142 (50%)	120 (53.1%)	262 (51.4%)	0.487
	Between 1st and 2nd Dose	20 (7%)	16 (7.1%)	36 (7.1%)	0.987
	After 2nd Dose	116 (40.8%)	83 (36.7%)	199 (39%)	0.343
	After 3rd Dose	6 (2.1%)	7 (3.1%)	133(2.5%)	0.483
^+^ Hospitalization	No	271 (95.8%)	216 (95.2%)	487 (95.5%)	0.743
	Yes	12 (4.2%)	11 (4.8%)	23 (4.5%)	
Infection in Family	No	53 (11%)	21 (6.9%)	74 (9.4%)	0.054
	Yes	429 (89%)	284 (93.1%)	713 (90.6%)	
Mortality in Family	No	273 (56.6%)	184 (60.3%)	457 (58.1%)	0.307
	Yes	209 (43.4%)	121 (39.7%)	330 (41.9%)	

Chi-squared test (*χ*^2^) was used with a significance level *Sig*. < 0.05. Statistically significant differences are indicated with bold character. ^+^ Respondents being COVID-19 infected (Yes).

**Table 4 vaccines-10-00621-t004:** COVID-19 Vaccine-related Anamnesis of Algerian Adults Participating in COVID-19 VBH Survey (*n* = 787).

Variable	Outcome	Non-Healthcare Professionals (*n* = 482)	Healthcare Professionals (*n* = 305)	Total (*n* = 787)	*Sig.*
Vaccine Type	Sinovac	334 (69.3%)	186 (61%)	520 (66.1%)	**0.016**
	Sinopharm	27 (5.6%)	15 (4.9%)	42 (5.3%)	0.678
	AstraZeneca-Oxford	66 (13.7%)	33 (10.8%)	99 (12.6%)	0.236
	Janssen	8 (1.7%)	16 (5.2%)	24 (3%)	**0.004**
	Sputnik V	27 (5.6%)	53 (17.4%)	80 (10.2%)	**<0.001**
	Pfizer-BioNTech	2 (0.4%)	2 (0.7%)	4 (0.5%)	0.643 *
	I do not know	18 (3.7%)	0 (0%)	18 (2.3%)	**<0.001**
Vaccine Technology	Inactivated Virus	361 (77.8%)	201 (65.9%)	562 (73.1%)	**<0.001**
	Adenoviral Vector	101 (21.8%)	102 (33.4%)	203 (26.4%)	**<0.001**
	mRNA-based	2 (0.4%)	2 (0.7%)	4 (0.5%)	0.672
Relief after Vaccination	Agree	287 (59.5%)	171 (56.1%)	458 (58.2%)	0.335
	Unsure	133 (27.6%)	87 (28.5%)	220 (28%)	0.777
	Disagree	62 (12.9%)	47 (15.4%)	109 (13.9%)	0.314
Prevention after Vaccination	No	55 (11.5%)	33 (10.8%)	88 (11.2%)	0.798
	Yes	427 (88.6%)	272 (89.2%)	699 (88.8%)	
Regret after Vaccination	Disagree	385 (79.9%)	232 (76.1%)	617 (78.4%)	0.206
	Unsure	47 (9.8%)	33 (10.8%)	80 (10.2%)	0.629
	Agree ^π^	50 (10.4%)	40 (13.1%)	90 (11.4%)	0.239
^π^ Reasons for Regret	Vaccines are not efficient	27 (5.6%)	26 (8.5%)	53 (6.7%)	0.111
	Post-vaccination infection	30 (6.2%)	24 (7.9%)	54 (6.9%)	0.374
	Post-vaccination side effects	10 (2.1%)	14 (4.6%)	24 (3%)	**0.046**
	Did not choose best vaccine	3 (0.6%)	3 (1%)	6 (0.8%)	0.682 *
	Disease became milder	1 (0.2%)	1 (0.3%)	2 (0.3%)	1.000 *
COVID-19 Vaccine Booster	No	420 (87.1%)	263 (86.2%)	683 (86.8%)	0.714
	Yes ^Ψ^	62 (12.9%)	42 (13.8%)	104 (13.2%)	
^Ψ^ Booster Dose Type	Sinovac	39 (50%)	21 (44.7%)	60 (48%)	0.564
	Sinopharm	3 (3.8%)	6 (12.8%)	9 (7.2%)	0.062
	AstraZeneca-Oxford	12 (15.4%)	4 (8.5%)	16 (12.8%)	0.265
	Janssen	7 (9%)	12 (25.5%)	19 (15.2%)	**0.013**
	Sputnik V	5 (6.4%)	1 (2.1%)	6 (4.8%)	0.278
	Pfizer-BioNTech	2 (2.6%)	3 (6.4%)	5 (4%)	0.291
	I do not know	10 (12.8%)	0 (0%)	10 (8%)	**0.010**

Chi-squared test (χ^2^) and Fisher’s exact test (*) were used with a significance level *Sig.* < 0.05. Statistically significant differences are indicated with bold character. ^π^ Respondents who are willing to receive COVID-19 vaccine booster dose. ^Ψ^ Respondents who received COVID-19 vaccine booster dose.

**Table 5 vaccines-10-00621-t005:** COVID-19 Vaccine Booster-related Attitudes of Algerian Adults Participating in COVID-19 VBH Survey (*n* = 787).

Variable	Outcome	Non-Healthcare Professionals (*n* = 482)	Healthcare Professionals (*n* = 305)	Total (*n* = 787)	*Sig.*
Willingness	Rejection ^‡^	95 (19.7%)	102 (33.4%)	197 (25%)	**<0.001**
	Hesitancy	121 (25.1%)	63 (20.7%)	184 (23.4%)	0.151
	Acceptance ^+^	266 (55.2%)	140 (45.9%)	406 (51.6%)	**0.011**
^+^ Reasons for Acceptance	There is no alternative	72 (15%)	37 (12.1%)	109 (13.9%)	0.262
	I want to travel abroad	64 (13.3%)	44 (14.4%)	108 (13.7%)	0.657
	Experts recommend it	141 (29.3%)	52 (17%)	193 (24.6%)	**<0.001**
	It is necessary and efficient	119 (24.7%)	65 (21.3%)	184 (23.4%)	0.269
^+^ Preferred Vaccine Type	Sinovac	93 (35%)	42 (30%)	135 (33.3%)	0.313
	Sinopharm	5 (1.9%)	3 (2.1%)	8 (2%)	1.000 *
	AstraZeneca-Oxford	31 (11.7%)	17 (12.1%)	48 (11.8%)	0.885
	Janssen	21 (7.9%)	30 (21.4%)	51 (12.6%)	**<0.001**
	Sputnik V	18 (6.8%)	11 (7.9%)	29 (7.1%)	0.685
	Pfizer-BioNTech	28 (10.5%)	11 (7.9%)	39 (9.6%)	0.386
	Moderna	6 (2.3%)	3 (2.1%)	9 (2.2%)	1.000 *
	No Preference	23 (8.6%)	7 (5%)	30 (7.4%)	0.182
^‡^ Reasons for Rejection	Primer doses are sufficient	72 (14.9%)	50 (16.4%)	122 (15.5%)	0.583
	Fear of side effects	18 (3.7%)	21 (6.9%)	39 (5%)	**0.047**
	Vaccination is inefficient	36 (7.5%)	27 (8.9%)	63 (8%)	0.486
	It can harm immune system	31 (6.4%)	20 (6.6%)	51 (6.5%)	0.944
	I had breakthrough infection	2 (0.4%)	4 (1.3%)	6 (0.8%)	0.214 *

Chi-squared test (*χ*^2^) and Fisher’s exact test (*) were used with a significance level *Sig.* < 0.05. Statistically significant differences are indicated with bold character. ^+^ Respondents who regret being vaccinated. ^‡^ Respondents who rejected the COVID-19 vaccine booster dose.

**Table 6 vaccines-10-00621-t006:** Demographic and Anamnestic Determinants of Vaccine Booster-related Attitudes among Algerian Adults Participating in COVID-19 VBH Survey (*n* = 787).

Variable	Outcome	Rejection (*n* = 197)	*Sig.*	Hesitancy (*n* = 184)	*Sig.*	Acceptance (*n* = 406)	*Sig.*
Sex	Female	138 (28.5%)	**0.005**	122 (25.2%)	0.136	225 (46.4%)	**<0.001**
	Male	59 (19.5%)		62 (20.5%)		181 (59.9%)	
Age Group	18–30 years old	64 (30.2%)	**0.043**	55 (25.9%)	0.302	93 (43.9%)	**0.008**
	31–40 years old	66 (26.8%)	0.432	56 (22.8%)	0.783	124 (50.4%)	0.655
	41–50 years old	45 (23.4%)	0.558	51 (26.6%)	0.231	96 (50%)	0.613
	51–60 years old	19 (19.4%)	0.168	14 (14.3%)	**0.023**	65 (66.3%)	**0.002**
	>60 years old	3 (7.7%)	**0.010**	8 (20.5%)	0.664	28 (71.8%)	**0.010**
Marital Status	Single	76 (24.8%)	0.920	79 (25.8%)	0.198	151 (49.3%)	0.315
	Married	121 (25.2%)		105 (21.8%)		255 (53%)	
Residence	Urban	181 (25.2%)	0.711	169 (23.5%)	0.736	368 (51.3%)	0.544
	Rural	16 (23.2%)		15 (21.7%)		38 (55.1%)	
Educational Level	High School	5 (7.8%)	**<0.001**	13 (20.3%)	0.545	46 (71.9%)	**<0.001**
	Bachelor’s Degree	98 (28.5%)	**0.049**	84 (24.4%)	0.544	162 (47.1%)	**0.026**
	Masters’ Degree or above	94 (24.8%)	0.886	87 (23%)	0.786	198 (52.2%)	0.723
Chronic Illness	Diabetes Mellitus	9 (14.3%)	**0.040**	14 (22.2%)	0.821	40 (63.5%)	**0.049**
	Chronic Hypertension	19 (27.1%)	0.669	9 (12.9%)	**0.029**	42 (60%)	0.140
	Cardiovascular Disease	6 (40%)	0.225 *	2 (13.3%)	0.540 *	7 (46.7%)	0.700
	Respiratory Disease	10 (22.7%)	0.716	6 (13.6%)	0.116	28 (63.6%)	0.100
	Renal Disease	1 (25%)	1.000 *	0 (0%)	0.578 *	3 (75%)	0.625 *
	Other	28 (29.2%)	0.318	24 (25%)	0.689	44 (45.8%)	0.229
	Total	49 (22.4%)	0.285	44 (20.1%)	0.176	126 (57.5%)	**0.038**
Influenza Vaccine	No	153 (25.7%)	0.465	146 (24.5%)	0.191	297 (49.8%)	0.082
	Yes	44 (23%)		38 (19.9%)		109 (57.1%)	
COVID-19 Infection	No	50 (18.3%)	**0.002**	69 (25.3%)	0.360	154 (56.4%)	**0.049**
	Yes ^+^	147 (28.6%)		115 (22.4%)		252 (49%)	
^+^ Onset	Before 1st Dose	76 (29%)	0.690	62 (23.7%)	0.536	124 (47.3%)	0.381
	Between 1st and2nd Dose	10 (27.8%)	0.950	12 (33.3%)	0.108	14 (38.9%)	0.199
	After 2nd Dose	58 (29.1%)	0.715	41 (20.6%)	0.400	100 (50.3%)	0.708
	After 3rd Dose	0 (0%)	**0.022**	0 (0%)	**0.047 ***	13 (100%)	**<0.001**
^+^ Hospitalization	No	141 (29%)	0.767	113 (23.2%)	0.104	233(47.8%)	0.103
	Yes	6 (26.1%)		2 (8.7%)		15 (65.2%)	
Infection in Family	No	12 (16.2%)	0.066	13 (17.6%)	0.215	49 (66.2%)	0.008
	Yes	185 (25.9%)		171 (24%)		357 (50.1%)	
Mortality in Family	No	115 (25.2%)	0.920	104 (22.8%)	0.627	238 (52.1%)	0.746
	Yes	82 (24.8%)		80 (24.2%)		168 (50.9%)	
Vaccine Type	Sinovac	137 (26.3%)	0.235	125 (24%)	0.542	258 (49.6%)	0.122
	Sinopharm	9 (21.4%)	0.580	9 (21.4%)	0.759	24 (57.1%)	0.459
	AstraZeneca-Oxford	22 (22.2%)	0.490	20 (20.2%)	0.424	57 (57.6%)	0.202
	Janssen	10 (41.7%)	0.056	7 (29.2%)	0.496	7 (29.2%)	**0.026**
	Sputnik V	15 (18.8%)	0.171	18 (22.5%)	0.844	47 (58.8%)	0.176
	Pfizer-BioNTech	0 (0%)	0.247	0 (0%)	0.578 *	4 (100%)	0.125
	I do not know	4 (22.2%)	0.781	5 (27.8%)	0.585 *	9 (50%)	0.891
Vaccine Technology	Inactivated Virus	146 (26%)	0.353	134 (23.8%)	0.540	282 (50.2%)	0.186
	Adenoviral Vector	47 (23.2%)	0.456	45 (22.2%)	0.663	111 (54.7%)	0.310
	mRNA-based	0 (0%)	0.246	0 (0%)	0.578 *	4 (100%)	0.052
Relief after Vaccination	Agree	59 (12.9%)	**<0.001**	91 (19.9%)	0.006	308 (67.2%)	**<0.001**
	Unsure	70 (31.8%)	**0.006**	74 (33.6%)	**<0.001**	76 (34.5%)	**<0.001**
	Disagree	68 (62.4%)	**<0.001**	19 (17.4%)	0.114	22 (20.2%)	**<0.001**
Prevention after Vaccination	No	26 (29.5%)	0.300	21 (23.9%)	0.909	41 (46.6%)	0.320
	Yes	171 (24.5%)		163 (23.3%)		365 (52.2%)	
Regret after Vaccination	Disagree	105 (17%)	**<0.001**	143 (23.2%)	0.797	369 (59.8%)	**<0.001**
	Unsure	28 (35%)	**0.030**	29 (36.3%)	**0.004**	23 (28.7%)	**<0.001**
	Agree ^‡^	64 (71.1%)	**<0.001**	12 (13.3%)	**0.017**	14 (15.6%)	**<0.001**
^‡^ Reasons for Regret	Vaccines are not efficient	36 (67.9%)	**<0.001**	6 (11.3%)	**0.032**	11 (20.8%)	**<0.001**
	Post-vaccination infection	34 (63%)	**<0.001**	10 (18.5%)	0.382	10 (18.5%)	**<0.001**
	Post-vaccination side effects	15 (62.5%)	**<0.001**	2 (8.3%)	0.077	7 (29.2%)	0.026
	Did not choose best vaccine	5 (83.3%)	**0.005** *	0 (0%)	0.345 *	1 (16.7%)	0.113 *
	Disease became milder	2 (100%)	0.062 *	0 (0%)	1.000 *	0 (0%)	0.234 *

Chi-squared test (*χ*^2^) and Fisher’s exact test (*) were used with a significance level *Sig*. < 0.05. Statistically significant differences are indicated with bold character. ^+^ Respondents being COVID-19 infected (Yes). ^‡^ Respondents who regret being vaccinated.

**Table 7 vaccines-10-00621-t007:** Regression Analysis of Demographic and Anamnestic Factors for COVID-19 VB Acceptance.

Predictor	B (SE)	Wald	OR	CI 95%	*Sig.*
Sex: Male (vs. Female)	0.547 (0.149)	13.565	1.729	1.292–2.313	**<0.001**
Age Group: 31–40 yo (vs. 18–30 yo)	0.263 (0.188)	1.950	1.301	0.899–1.881	0.163
Age Group: 41–50 yo (vs. 18–30 yo)	0.247 (0.200)	1.520	1.280	0.865–1.894	0.218
Age Group: 51–60 yo (vs. 18–30 yo)	0.924 (0.255)	13.178	2.520	1.530–4.152	**<0.001**
Age Group: >60 yo (vs. 18–30 yo)	1.181 (0.382)	9.565	3.257	1.541–6.884	**0.002**
Education: BA./BSc. (vs. College/School)	−1.055 (0.298)	12.504	0.348	0.194–0.625	**<0.001**
Education: MSc. or above (vs. College/School)	−0.848 (0.296)	8.193	0.428	0.239–0.765	**0.004**
Profession: Healthcare (vs. Non-healthcare)	−0.373 (0.147)	6.427	0.689	0.517–0.919	**0.011**
Chronic Illness: Yes (vs. No)	0.332 (0.160)	4.280	1.394	1.018–1.908	**0.039**
COVID-19 Infection: No (vs. Yes)	0.297 (0.151)	3.882	1.345	1.002–1.807	**0.049**
Post-vaccination Relief: Agree (vs. Disagree)	2.094 (0.259)	65.601	8.120	4.892–13.479	**<0.001**
Post-vaccination Regret: Disagree (vs. Agree)	2.089 (0.302)	47.785	8.077	4.467–14.605	**<0.001**

Binary logistic regression was used with a significance level *Sig*. < 0.05. Statistically significant differences are indicated with bold character.

**Table 8 vaccines-10-00621-t008:** Regression Analysis of Psychological Promoters of and Barriers to COVID-19 VB Acceptance.

Predictor	B (SE)	Wald	AOR	CI 95%	*Sig.*
There is no alternative: Agree (vs. Disagree)	1.180 (0.253)	21.824	3.256	1.984–5.342	**<0.001**
I want to travel abroad: Agree (vs. Disagree)	0.590 (0.236)	6.261	1.804	1.136–2.863	**0.012**
Experts recommend it: Agree (vs. Disagree)	1.569 (0.221)	50.570	4.801	3.116–7.398	**<0.001**
It is necessary and efficient: Agree (vs. Disagree)	3.336 (0.384)	75.348	28.112	13.235–59.709	**<0.001**
Primer doses are sufficient: Disagree (vs. Agree)	3.163 (0.386)	67.029	23.641	11.087–50.409	**<0.001**
Fear of side effects: Disagree (vs. Agree)	0.391 (0.500)	0.612	1.479	0.555–3.943	0.434
Vaccination is inefficient: Disagree (vs. Agree)	1.641 (0.471)	12.151	5.159	2.051–12.979	**<0.001**
It can harm immune system: Disagree (vs. Agree)	1.612 (0.516)	9.776	5.013	1.825–13.770	**0.002**
I had breakthrough infection: Disagree (vs. Agree)	1.927 (1.108)	3.026	6.870	0.783–60.248	0.082

Multivariable logistic regression was adjusted for sex, age group, educational level, profession, chronic illness, previous COVID-19 infection, post-vaccination relief and regret. Statistically significant differences are indicated with bold character.

## Data Availability

The data that support the findings of this study are available from the corresponding author upon reasonable request.

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
