# Peer review of "COVID-19 Vaccine Booster Hesitancy (VBH) and Its Drivers in Algeria: National Cross-Sectional Survey-Based Study"

_vaccines, 2022, doi:10.3390/vaccines10040621_

Round 1

Reviewer 1 Report

In this article, researchers presented the result of an online survey evaluating the perception of people on COVID-19 vaccine booster dose in Algeria. This is a timely and well-presented research. It shows a low level of vaccine booster dose acceptance in Algeria and several factors which serve as the barrier and enabler of vaccine acceptance. This can be helpful in planning further population education. I would like to suggest changing the title and removing the word prevalence. Likewise, I suggest inclusion of further information about COVID-19, its vaccination status, and vaccine hesitancy problem in general in Algeria early on during introduction. 

Reviewer 2 Report

I was invited to revise the paper entitled "Prevalence and Drivers of COVID-19 Vaccine Booster Hesitancy (VBH) in Algeria: National Cross-sectional Survey-based Study". It was a cross-sectional study that aimed to evaluate the vaccine booster hesitancy and acceptance in Algeria's and to evaluate associated factors. The topic is interesting and focused on a importante aspect of covid vaccination campaign. In addition, few similar studies were performed in low income countries.

Major observations:

  • In the introduction section, Authors should describe how the vaccination campaign was structured in Algeria and which vaccines were available;
  • Participants were invited using social media platforms, so there is a selection bias. Subjects that did not have access to social networks are unreachable;
  • Itis unknown why Authors performed a comparison between HCW and non-HCW. This is not the main aim of the study. However it is an interesting supplementary analysis; Authors should permorm mainly the analysis in the general sample; Probably HCW more likely accept booster dose;
  • Tests reported in tables 3,4,5 amd 6 should be corrected for multiple comparisons;
  • It is unknown if multivariate analysis was a fully adjusted model. Authors should be more clear;
  • Authors should avoid the use of the word "Predictor" related t ologistic regression. This is a cross-sectional study!
  • Discussion section should compare their results with previous literature.

Reviewer 3 Report

The study aim is important. Data from Africa are really needed, due to the global dimension of the COVID-19 pandemic.

This study is interesting, but some revisions should be applied.

  1. In the introduction section - please provide a new paragraph on COVID-19 in Africa, especially Algeria. A number of cases, dynamics of the COVID-19 pandemic as well as basic data on COVID-19 vaccination program (including organization and availability of the vaccines). Moreover, data on vaccines available in Algeria should be clearly presented. This is particularly important for international readers. 
  2. The limitations of the study design should be discussed at the end of the discussion section.
  3. In the results section - please provide data on the response rate (if available)
  4. The study group was divided into medical and non-medical. Please justify this approach. Moreover, how does it match with the study title: National Cross-sectional Survey-based 
  5. The Authors used "CoronaVac" - the name of the vaccine, but in the case of other vaccines, the names of the company were mentioned: Janssen and AstraZeneca. Please unify and provide equal naming for each of the vaccines (name of products or name of company).
  6. The limitations section should be more precise. The current version is to limit (e.g. study design etc.)  
  7. Please add 2-3 sentences on the practical implications of this study, especially for the African region. 
  8. Please revise the conclusions section. The current version repeats results, rather than presenting clear conclusions. 
  9. The manuscript should be proofread by an English native speaker.

Round 2

Reviewer 2 Report

Authors addressed all points raised. It can be accepted for publication

Reviewer 3 Report

This manuscript was significantly improved. The Author applied all the comments. This manuscript may have a significant impact on better understanding the African perspective on the COVID-19 pandemic. I would like to congratulate the Authors. This is a really interesting and important manuscript.